# A High-Performance Magnetic Shield with MnZn Ferrite and Mu-Metal Film Combination for Atomic Sensors

**DOI:** 10.3390/ma15196680

**Published:** 2022-09-26

**Authors:** Xiujie Fang, Danyue Ma, Bowen Sun, Xueping Xu, Wei Quan, Zhisong Xiao, Yueyang Zhai

**Affiliations:** 1School of Physics, Beihang University, Beijing 100191, China; 2Zhejiang Provincial Key Laboratory of Ultra-Weak Magnetic-Field Space and Applied Technology, Hangzhou Innovation Institute of Beihang University, Hangzhou 310000, China; 3Key Laboratory of Ultra-Weak Magnetic Field Measurement Technology, Ministry of Education, School of Instrumentation and Optoelectronic Engineering, Beihang University, Beijing 100191, China

**Keywords:** MnZn ferrite, mu-metal film, magnetic shield, magnetic noise, atomic sensors

## Abstract

This study proposes a high-performance magnetic shielding structure composed of MnZn ferrite and mu-metal film. The use of the mu-metal film with a high magnetic permeability restrains the decrease in the magnetic shielding coefficient caused by the magnetic leakage between the gap of magnetic annuli. The 0.1–0.5 mm thickness of mu-metal film prevents the increase of magnetic noise of composite structure. The finite element simulation results show that the magnetic shielding coefficient and magnetic noise are almost unchanged with the increase in the gap width. Compared with conventional ferrite magnetic shields with multiple annuli structures under the gap width of 0.5 mm, the radial shielding coefficient increases by 13.2%, and the magnetic noise decreases by 21%. The axial shielding coefficient increases by 22.3 times. Experiments verify the simulation results of the shielding coefficient of the combined magnetic shield. The shielding coefficient of the combined magnetic shield is 16.5%. It is 91.3% higher than the conventional ferrite magnetic shield. The main difference is observed between the actual and simulated relative permeability of mu-metal films. The combined magnetic shielding proposed in this study is of great significance to further promote the performance of atomic sensors sensitive to magnetic field.

## 1. Introduction

Numerous ultra-high sensitivity atomic sensors that use quantum effects to detect physical quantities, such as atomic magnetometer [1,2], atomic gyroscope [3,4], atomic clock [5], and superconducting quantum interferometer [6,7], are magnetic-field sensitive. The stability of the environmental magnetic field has a direct effect on the measurement performance of sensors. Consequently, passive magnetic shielding and active magnetic compensation are widely used in the aforementioned sectors [8,9,10].

The accuracy of the magnetometer limits the accuracy of the active magnetic compensation. Moreover, its high-power consumption requires an external current source [9,10]. The passive magnetic shield uses the characteristics of higher permeability of magnetic material compared with air to cancel the magnetic flux density around the material. Utilizing a magnetic material with a higher permeability sufficiently reduces the magnetic flux density within the magnetic shield. A common magnetic shield that achieves a high shielding coefficient involves using a mu-metal multilayer shielding [11]. This type of magnetic shield successfully suppresses the influence of the fluctuating magnetic field in the environment on the sensors inside the magnetic shield. Nevertheless, it possesses a high conductivity. The Johnson current generated by the material results in a magnetic noise level of ~10 fT/Hz^1/2^. This prevents the atomic sensors from being made more sensitive. A high-performance magnetic shield must have a high shielding coefficient and a low magnetic noise [12,13].

Reducing the magnetic noise of magnetic shielding systems has become a research priority in recent years. Kornack and Lee investigated the analytical formula for magnetic shielding noise [14,15]. The magnetic noise was computed using the power loss based on the fluctuation dissipation theory [16]. It was discovered that the magnetic noise was mostly caused by the complex permeability and conductivity of the material. The conductivity of MnZn ferrite material was ~10^−6^ S/m less than that of mu-metal material [17,18]. Through theoretical calculation and experimental research, the noise of the ferrite shield has been reduced by 25 times to 0.75 fT/Hz^1/2^ compared with the noise of the mu-metal shield, which is above 40 Hz. Bevan employed square ferrite magnetic shielding in 2018 to reduce the magnetic noise level of the NMR gyroscope [19]. In addition, low-noise ferrite shielding is utilized in several high-sensitivity atomic sensors and measurement techniques, such as co-magnetometer [20,21], demonstrating its superior performance.

As observed in the previous research, the practical application of a ferrite magnetic shield presents several challenges. In lieu of being integrally created, numerous magnetic annuli are spliced together to produce the ferrite magnetic shield to meet the requirements of a large-scale magnetic shielding system and limit the influence of magnetic noise [22,23]. Due to the surface roughness and non-uniformity of the magnetic annuli, there will be an air gap, approximately 0.1–0.5 mm, in the ferrite shield when magnetic annuli are spliced [22]. The air gap between magnetic annuli diminishes the shielding coefficient and increases the radial magnetic noise. The shielding coefficient of ferrite magnetic shielding deteriorates. This decreases the environmental magnetic noise reduction and increases self-induced magnetic noise. Its effectiveness is impacted by the superposition of the two forms of magnetic noise. The research on the air gap of ferrite magnetic annuli focuses solely on its effect on the shielding coefficient and magnetic noise. There is currently no solution for the air gap. The air gap cannot be eliminated by finishing the contact surface of magnetic annuli.

This research quantitatively evaluates how the air gap of ferrite annuli affects the axial and radial shielding coefficient and magnetic noise. To reduce the effect of the gap on the shielding coefficient and magnetic noise, a magnetic shielding structure comprised of ferrite and mu-metal film is proposed. This configuration can use the mu-metal material to suppress the magnetic leakage field at the ferrite air gap. Outside the ferrite magnetic shielding, a mu-metal film is adhered to lessen the magnetic shielding noise of the mu-metal film. The shielding performance of ferrite and mu-metal film combination magnetic shield (FMCS) is quantified using the finite element method (FEM). This configuration has a better shielding coefficient than conventional ferrite magnetic shields. The magnetic noise study results indicated that this design reduced the effect of the air gap on magnetic noise. In addition, the configuration accounted for the occurrence of an air gap between the ferrite magnetic shield and the mu-metal film when they adhered. In this case, the shielding coefficient and magnetic noise were also explored in this work. Finally, the magnetic shielding coefficient measuring platform was constructed, and the experiment verified the accuracy of the calculation.

## 2. Methods

### 2.1. The Magnetic Shielding Structure Composed of Ferrite and Mu-Metal Film

The FMCS schematic comprises a ferrite shield and a mu-metal film. In Figure 1, the *x*-direction is the radial, while the *z*-direction is the axial. The size of the magnetic shielding model established by simulation and experiment is identical. Five magnetic annuli and two end caps constitute the ferrite shield. Each annulus has a height of 45 mm, an inner diameter of 114 mm, and a wall thickness of 13 mm. The thickness of the two end caps is 10 mm. In practical application, there are four 22 mm diameter access holes in the radial direction and two 28 mm diameter access holes in the axial direction. This study exclusively examines the effect of the air gap between ferrite magnetic annuli to prevent the access hole from influencing the results, and the access hole is not accounted for in the simulation. The ferrite magnetic shield material is the soft MnZn ferrite with a spinel structure. Figure 2 depicts the morphology of the MnZn ferrites, consisting of aggregated crystallites with homogeneous grains. Low porosity is present in the MnZn ferrite sample, and no air gaps emerge at the grain boundary. Owing to the discontinuous grain development during the sintering process, a few vacancies are observed within the grain. This intra-granular vacancy may impede the mobility of the domain wall, resulting in an adverse effect on the magnetic permeability. The composition of the MnZn ferrite is analyzed using an inductively coupled plasma emission spectrometer (Agilent, Palo Alto, CA, USA). The main elements, Fe, Mn, and Zn are shown in Table 1, in which other elements such as Na, P, Si, and Ca are also included. The addition of trace elements can reduce the loss. However, the mass fraction (wt.%) of other elements should not exceed 0.15%. The complex permeability of the ferrite magnetic shield is also measured experimentally. The real component of the complex permeability is 10029, and the imaginary component is 150 [24]. 

A mu-metal film is adhered to the exterior of the ferrite magnetic shield. The thickness of the film ranges from 0.1 mm to 0.5 mm. The real component of the complex permeability of mu-metal film is 30,000, while the imaginary component is 1000. There are two methods for pasting. The first one involved tightly pasting the ferrite and mu-metal film, and the outside of the film was fixated using adhesive tape. The second one involved attaching the ferrite and mu-metal film using an insulating double-sided tape, creating a space between them. Both cases were analyzed in this work.

### 2.2. Analysis Method of Shielding Coefficient and Noise

The magnetic shielding coefficient is represented by the coefficient *S*. It is defined as the ratio of the magnetic flux density *B*_shield_ at the center point of the magnetic shield when the shield was present to the magnetic flux density *B_0_*, with the shield at the same point without the shield. The equation is as follows [25]:(1)$$S=\frac{B_{\mathrm{shield}}}{B_{0}}$$

The magnetic shielding coefficient for complex structures can be calculated using FEM and the commercial program ANSYS Electromagnetics Suite 19.2 (ANSYS Maxwell 3D, Canonsburg, PA, USA). To generate a highly uniform space magnetic field during the simulation of the static shielding coefficient, the magnetic field was simulated using the boundary condition approach. During the simulation, the magnetic flux density used was 25 μT, which was the actual measured magnetic flux density of the environment. Magnetic noise was an additional essential parameter for evaluating the performance of magnetic shielding. It consisted of the residual environmental magnetic noise after magnetic shielding and magnetic noise produced by magnetic shielding materials. The multilayer magnetic shield could attain a shielding coefficient of 10^4^ to 10^6^. The residual ambient magnetic noise was less than 0.1 fT/Hz^1/2^, and the magnetic noise generated by magnetic shielding materials dominated.

Generally, reciprocal approaches were used to calculate the magnetic noise *δB*. The magnetic noise was obtained by calculating the power loss, *P*_s_*,* in the material. *P*_s_ was caused by the magnetic field generated by the known excitation coil. The area was *A*, and the carry current is *I* [14,15].
(2)$$\delta B=\frac{\sqrt{8k_{B}T}\sqrt{P_{s}}}{\omega AI}$$
where the Boltzmann constant is *k*_B_, angular frequency is *ω,* and the Kelvin temperature is *T*. The power loss of low-frequency materials was mainly composed of eddy current power loss $P_{e}=\int_{V}\frac{1}{2}\sigma E^{2}dV$ and hysteresis power loss $P_{h}=\int_{V}\frac{1}{2}\mu^{\prime \prime }H_{m}{}^{2}dV$. The magnetic noise of FMCS is:(3)$$\delta B_{\mathrm{FBCS}}=\frac{\sqrt{8k_{B}T}\sqrt{P_{{}_{e}}^{tot}+P_{{}_{h}}^{tot}+P_{{}_{e}}^{mu}+P_{{}_{h}}^{mu}}}{\omega AI}$$
where $P_{{}_{e}}^{tot}$ and $P_{{}_{h}}^{tot}$ are the eddy current power loss and hysteresis power loss of ferrite shield, respectively. $P_{{}_{e}}^{mu}$ and $P_{{}_{h}}^{mu}$ are the eddy current power loss and hysteresis power loss of mu-metal film, respectively.

## 3. Experimental Calculation Results and Discussion

### 3.1. Magnetic Shielding Coefficient

A Ferrite magnetic shield is difficult to manufacture and process, especially when a large size is required. For this reason, a cylindrical ferrite shield is usually made of several short ferrite annuli pasted with ceramic adhesive. The thickness of the ceramic adhesive and the unevenness of the contact surface led to air gaps [23]. The air gap between magnetic annuli causes magnetic leakage. The cloud diagram of magnetic flux density when the air gap width was 0.5 mm is shown in Figure 3a, and Figure 3b demonstrates that the FMCS structure effectively prevented magnetic leakage at the air gap of the ferrite magnetic shield. The thickness of mu-metal film used for simulation was 0.5 mm.

The relationship between mu-metal film thickness and magnetic shielding coefficient was investigated to quantify the inhibitory effect of FMCS structure on the air gap of the ferrite shield and to enhance the shielding coefficient. The conventional ferrite magnetic shielding coefficient variation with the gap width was first calculated. As shown in Figure 4, the *x*- and *z*-direction shielding coefficients decreased by 16.2% and 98.1%, respectively, when the gap width changed from 0 to 0.5 mm.

Figure 5 illustrates the relationship between the magnetic shielding coefficient of the FMCS structure and the gap width for different film thicknesses. As shown in Figure 5a, the *x*-direction shielding coefficient of FMCS structure with varying film thicknesses decreases gradually as the gap width increases. When gap width increased from 0 to 0.5 mm, the shielding coefficient decreased by approximately 3.8%. Shielding coefficient increased as layer thickness increased. When the gap width was equivalent to 0.5 mm, the shielding coefficient of film (0.1 mm and 0.5 mm) increased by 9.8% and 13.2%, respectively, compared with conventional ferrite shielding. FMCS structure effectively suppressed the influence of the air gap on the *x*-direction shielding coefficient of conventional ferrite magnetic shielding. Figure 5b shows the relationship between the *z*-direction magnetic shielding coefficient of the FMCS structure and the width of the gap for different film thicknesses. Although the *z*-direction shielding coefficient decreased in the presence of an air gap, the width of the gap did not affect the shielding coefficient. In addition, the shielding coefficient improved with an increase in the layer thickness. When the gap width was 0.5 mm and film thickness was 0.1 mm, the *z*-direction shielding coefficient of the FMCS structure was 350, whereas the traditional ferrite magnetic shielding coefficient was only 15. The shielding coefficient increased by a factor of 22.3.

In addition, the ferrite and mu-metal film were attached to the FMCS framework with insulating double-sided tape, resulting in a gap between them. This configuration was also examined. Figure 6 depicts the results of the shielding coefficient calculation. When there was an air gap (width = 0.1 mm) between ferrite and mu-metal film (thickness = 0.5 mm), the shielding coefficient decreased by approximately 4.1% and 76.2%, respectively, compared to that in close contact. In practical application, ferrite and mu-metal films must be firmly pasted.

### 3.2. Magnetic Shield Noise

Magnetic noise is another important parameter for evaluating the shielding performance. Magnetic noise limits the enhancement of the sensitivity index, particularly in ultra-high-sensitivity magnetometers [26]. The sensitive axis of the magnetic field is generally oriented in the radial direction with an excellent magnetic shielding coefficient [2,4,27]. This study analyzed the influence of air gap on the shielding coefficient of conventional ferrite magnetic shields and FMCS structures. As shown in Figure 7, the air gap width increased from 0 to 0.5 mm, the thermal magnetization noise increased by 34.3%, and the eddy current noise decreased. However, the eddy current noise accounted for less than 1% of the total noise, which can be ignored.

The magnetic noise calculation results of the FMCS structure are shown in Figure 8. As illustrated in Figure 8a, when mu-metal film was close to the ferrite magnetic shield, the FMCS structure demonstrated a noticeable effect on suppressing the air gap of the ferrite magnetic annuli. The magnetic noise increased by 4.5% when the air gap width was extended from 0 to 0.5 mm, but the magnetic noise did not increase with the increase of air gap width. When the air gap width was 0.5 mm and the film thickness was 0.1 mm, the magnetic noise of the FMCS structure was 21% less than that of the typical ferrite magnetic shield. The magnetic noise could be reduced further by increasing the film thickness. Figure 8b shows the magnetic noise of the FMCS structure when the ferrite shield and the film were adhered with double-sided tape and there was a gap between them. In this case, despite being able to maintain the air gap, the magnetic noise reduction was only 6.3% less than with standard ferrite magnetic shielding. Increasing the thickness of the film could also reduce the magnetic noise, but the effect was minimal.

## 4. Experimental Setup and Results

As the current commercial atomic magnetometer has a sensitivity of tens of fT, it is difficult to detect the magnetic noise of ferrite magnetic shielding. The noise may be measured by building an ultra-high sensitive magnetic shielding measurement device based on the spin-exchange relaxation-free (SERF) effect. The device is complicated and is easily affected by optical or electrical noise detection. It is challenging to isolate the magnetic noise. Therefore, this study only measured the shielding coefficient to verify the advantages of the proposed FMCS structure and the simulation results. The schematic illustration of the magnetic shielding coefficient measurement platform is shown in Figure 9. The magnetic shield composed of one layer of aluminum magnetic shield and four layers of mu-metal magnetic shields suppressed the external magnetic field to avoid the fluctuation of the external magnetic field from distorting the measurement results. The combined magnetic shield had a shielding coefficient of approximately 10^5^ and a residual magnetic field of approximately 0.5 nT. The triaxial coil generated a known uniformly stable magnetic field. The magnetic field generated by the coil was enhanced under the impact of the ferromagnetic boundary of the outer combined magnetic shield. The three-axis coil was calibrated within the combined magnetic shield. The radial and axial coils were 25 nT/mA and 35 nT/mA, respectively. A commercial atomic magnetometer was used to measure the magnetic field in the FCMS structure.

For MnZn ferrite materials, *σ* ≈ 1 Ω^−1^m^−1^, correspondingly, the threshold frequency was higher than kHz, and below the threshold frequency, the shielding coefficient of ferrite magnetic shield did not change with the frequency, while the application frequency of SERF magnetometer was below 100 Hz. We also measured the frequency dependence of the shielding coefficient of the FMCS structure from DC to 100 Hz. The change of the shielding coefficient with frequency was only 2%. Figure 10 depicts the residual magnetic field in the FMCS structure and the conventional magnetic shield when the known DC external magnetic field was applied. Due to the increase in magnetic permeability, the *x*-direction shielding coefficient (a) of the conventional ferrite magnetic shield increased with the external magnetic field. When the external magnetic field was between 400 and 1800 nT, the average shielding coefficient was 703.26, which was close to the simulation value, proving that the analysis was accurate. The external magnetic field applied when measuring the *z*-direction magnetic shielding coefficient of conventional magnetic shielding was between 50 and 120 nT. The reason why the magnetic field applied in the *z*-direction was smaller than that in the *x*-direction was to prevent the residual magnetic field inside the shield from exceeding the measuring range of the atomic magnetometer due to the smaller measuring range of the atomic magnetometer and the smaller shielding coefficient in the *z*-direction. The average value of the *z*-direction shielding coefficient was 36.3, which confirmed that the air gap of a conventional ferrite magnetic shield with multiple annuli significantly impacted the axial shielding coefficient. Our experiments proved that the FMCS structures were superior to the conventional ferrite magnetic shielding. The average shielding coefficients of the FMCS structure (c) in the *x* and *z* directions were 819.22 and 383.6, respectively, which was an increase of 16.5% and 91.3% over the conventional ferrite magnetic shielding. The difference between the measured value and the simulated value was mainly due to the air gap during the film pasting process and the permeability of the material.

## 5. Conclusions

In this study, a high-performance magnetic shielding structure composed of ferrite and mu-metal film is proposed. Mu-metal film with a high relative permeability was used to suppress the reduction of magnetic shielding coefficient resulting from magnetic leakage in the magnetic annuli gap. The thickness of the mu-metal film, between 0.1 and 0.5 mm, prevented the increase in combined magnetic noise. FEM was used to calculate the shielding coefficient and magnetic noise of ferrite and mu-metal film combined shielding layer under varying gap widths of magnetic annuli. The results showed that the magnetic shielding coefficient and magnetic noise of the FMCS structures barely changed as the gap width increased. Under the gap width of 0.5 mm, the *x*-direction shielding coefficient increased by 13.2%, whereas the magnetic noise decreased by 21% compared to the conventional ferrite magnetic shielding with a gap. The shielding coefficient in the *z*-direction increased by 22.3 times. In addition, the influence of different assembly methods of ferrite and mu-metal film on the magnetic shielding performance was analyzed. The shielding performance declined when there was a gap between ferrite and the double-sided adhesive tape-pasted film. The gap between the film and the ferrite magnetic shielding must be avoided when the FMCS structure was employed to increase the shielding coefficient and decrease the magnetic noise. Amorphous, nanocrystalline, and other thin film materials may be utilized apart from mu-metal materials.

The experiments validated the shielding coefficient simulation results for the combined magnetic shield. The experimental results showed that the shielding coefficient of the composite magnetic shield was 16.5% and 91.3% greater than that of the conventional ferrite magnetic shield, respectively. The main difference was between the actual relative permeability of the mu-metal film and its simulated relative permeability. The high-performance combined magnetic shielding proposed in this study is of great significance for enhancing the sensitivity of atomic sensors.

## Figures and Tables

**Figure 1 materials-15-06680-f001:**
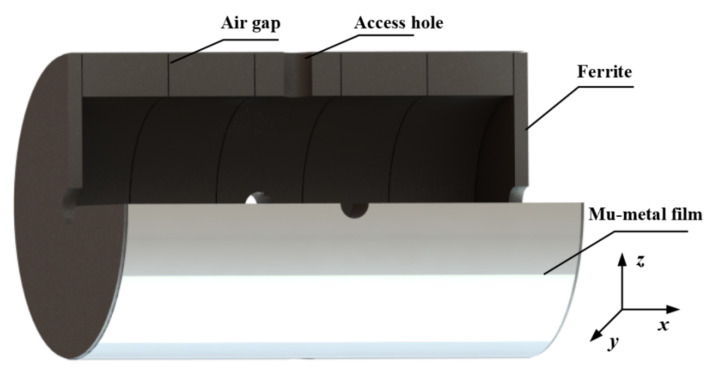
The schematic diagram of the FMCS structure.

**Figure 2 materials-15-06680-f002:**
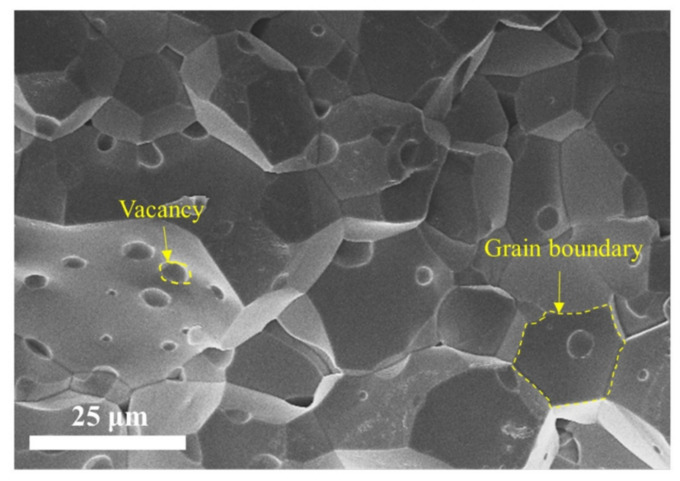
Morphology of high permeability MnZn ferrite. Grain boundary and air vacancy were indicated within the dashed lines.

**Figure 3 materials-15-06680-f003:**
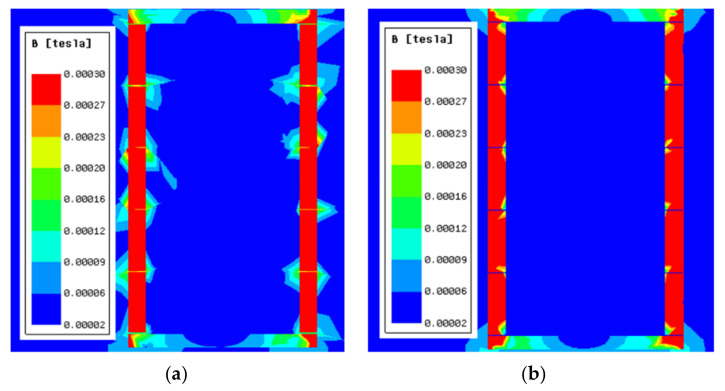
Magnetic flux density maps along the *x*–*z* plane; (**a**) conventional ferrite shield and (**b**) FMCS.

**Figure 4 materials-15-06680-f004:**
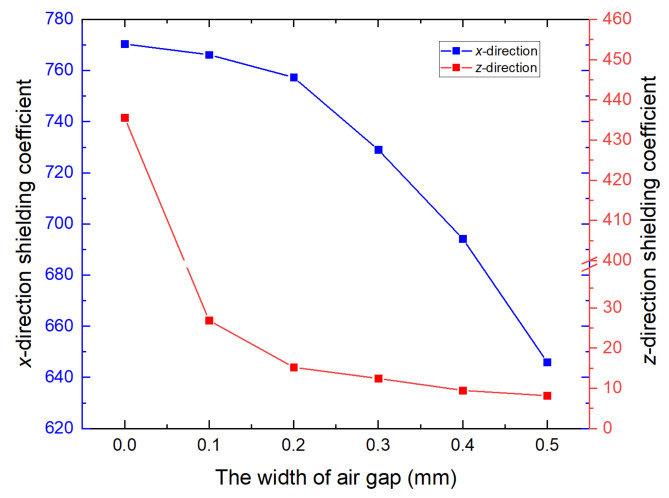
Relationship between magnetic shielding coefficient of the conventional ferrite and the gap width. When the gap width increases from 0 mm to 0.5 mm, the shielding coefficient for the *x*- and *z*-direction gradually decreases. The blue and red lines denote the *x-* and *z*-direction shielding coefficients, respectively.

**Figure 5 materials-15-06680-f005:**
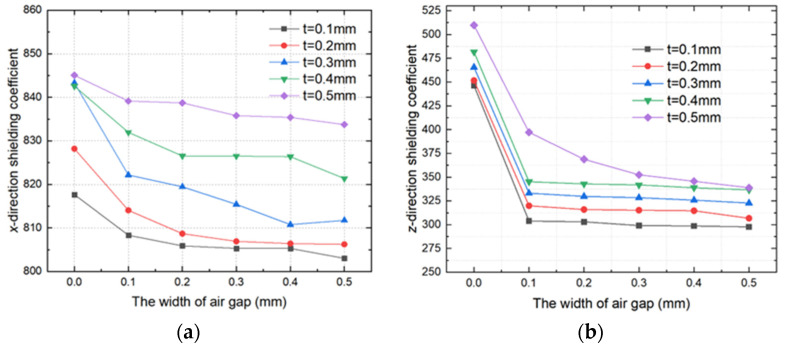
Relationship between the FMCS structure magnetic shielding coefficient and the gap width for different film thickness; (**a**) *x*-direction magnetic shielding coefficient, (**b**) *z*-direction magnetic shielding coefficient.

**Figure 6 materials-15-06680-f006:**
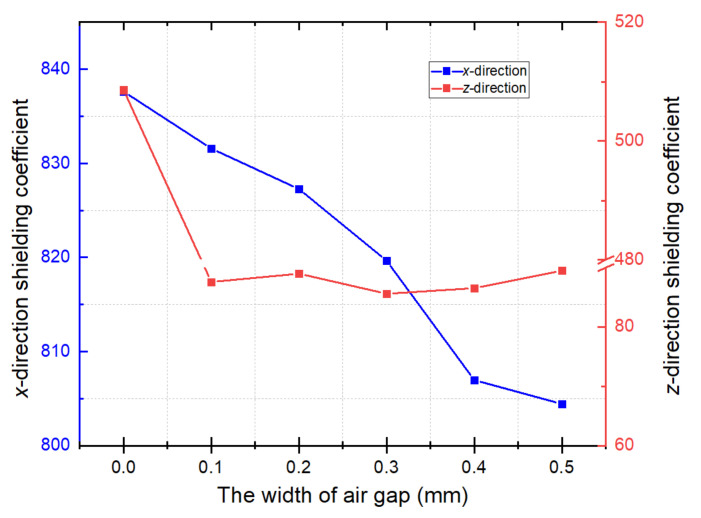
Shielding coefficient with an air gap between ferrite and mu-metal film. The ferrite and mu-metal film are fixed with insulating double-sided tape. The gap width caused by the isolation of double-sided adhesive tape is 0.1 mm.

**Figure 7 materials-15-06680-f007:**
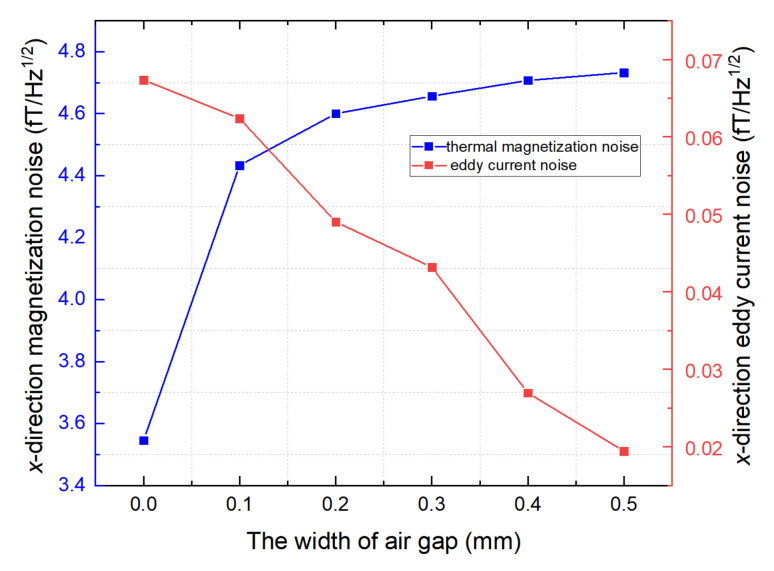
Magnetic noise of conventional ferrite magnetic shield in the presence of an air gap between ferrite annuli.

**Figure 8 materials-15-06680-f008:**
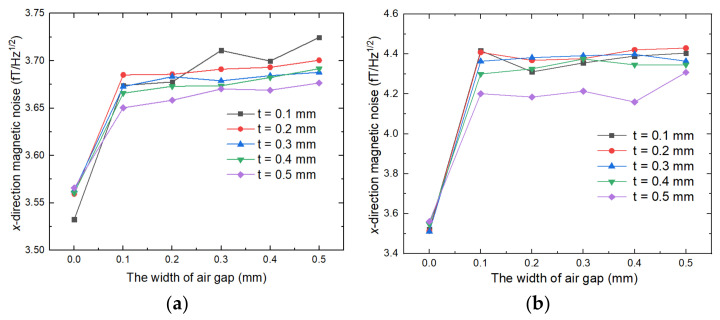
The magnetic noise calculation results of FMCS structure: (**a**) magnetic noise when there is no air gap between ferrite and mu-metal film and (**b**) magnetic noise when there is an air gap between ferrite and mu-metal film. The ferrite and mu-tal film are fixed with insulating double-sided tape.

**Figure 9 materials-15-06680-f009:**
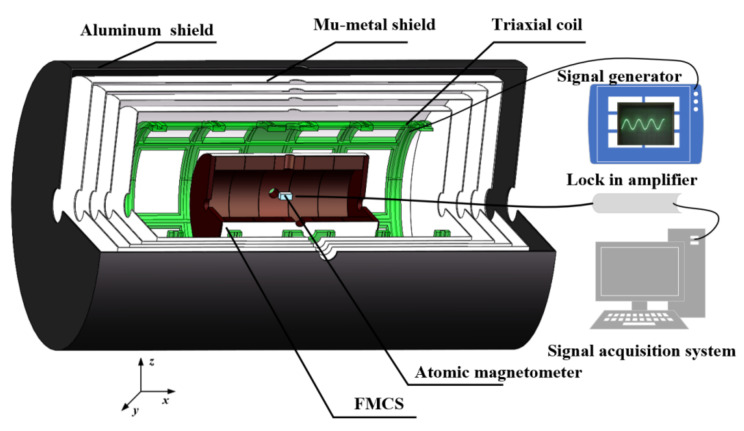
Schematic illustration of the magnetic shielding coefficient measurement platform. The ferrite shield size is the same as the structure used in the simulation, and the thickness of the film is 0.5 mm.

**Figure 10 materials-15-06680-f010:**
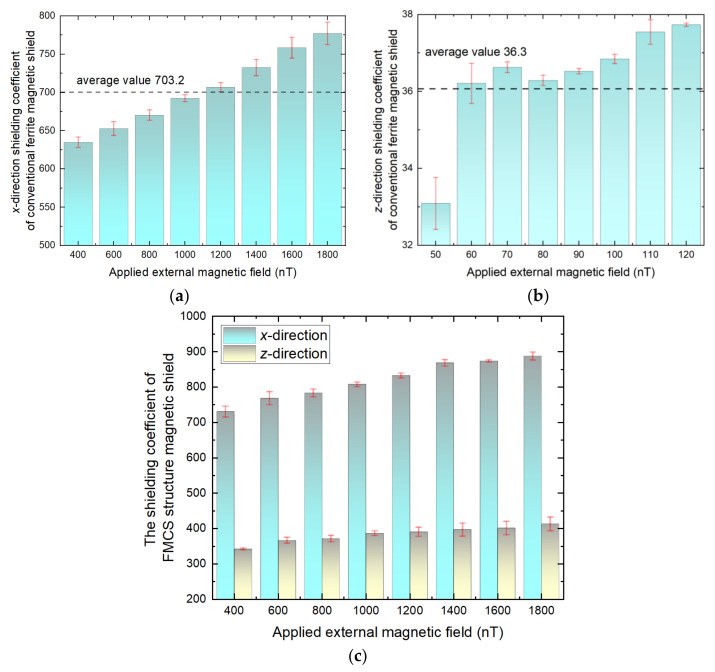
The (**a**) *x*-direction and (**b**) *z*-direction shielding coefficient of conventional ferrite magnetic shield versus applied external magnetic field; (**c**) Shielding coefficient of the FMCS when the external magnetic field is applied.

**Table 1 materials-15-06680-t001:** Composition analysis results of MnZn ferrite materials.

Element	Fe	Zn	Mn	Na	P	Si	Ca
wt.%	48.7568	13.4317	11.7299	0.0681	0.0676	0.0473	0.0288

## Data Availability

Not applicable.

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
