# Peer review of "A High-Performance Magnetic Shield with MnZn Ferrite and Mu-Metal Film Combination for Atomic Sensors"

_materials, 2022, doi:10.3390/ma15196680_

Round 1

Reviewer 1 Report

The work was done at a high level and is of interest to a wide scientific community.

comment:

Line 166: Figure 4 should be written instead of Figure 3

Author Response

We would first like to thank the reviewers for carefully reading our paper and giving us the opportunity to improve its quality. The revisions have been highlighted in the manuscript according to the requirement. Synchronous modifications and explanations have been carried out in the revised manuscript.

Point 1: Line 166: Figure 4 should be written instead of Figure 3

Response 1: We thank for the reviewer’s comments. According to your suggestion, we have corrected the error. Figure 4 is written instead of Figure 3.

Page 5 Line 166: As shown in Figure 4, the x- and z-direction shielding coefficients decreased by 16.2% and 98.1%, respectively, when the gap width changed from 0 to 0.5 mm.

Reviewer 2 Report

Dear Authors,

Please find attached the PDF file with comments as sticky notes.

Author Response

We would first like to thank the reviewers for carefully reading our paper and giving us the opportunity to improve its quality. The revisions have been highlighted in the manuscript according to the requirement. Synchronous modifications and explanations have been carried out in the revised manuscript.

Point 1: In Abstract: Axial direction and radial direction should be written instead of z- direction and x-direction. What are atomic sensors?

Response 1: We thank for the reviewer’s comments. According to your suggestion, axial direction and radial direction are written instead of z- direction and x-direction. As for atomic sensors, they mainly refer to sensors that use quantum effects to detect physical quantities, such as atomic magnetometer, atomic gyroscope, atomic clock, and superconducting quantum interferometer. Our proposed combined magnetic shielding is of great significance to further promote the performance of atomic sensors sensitive to magnetic field. We explained in the manuscript.

Page 1 Line 24: The combined magnetic shielding proposed in this study is of great significance to further pro-mote the performance of atomic sensors sensitive to magnetic field.

Page 1 Line 29: Numerous ultra-high sensitivity atomic sensors that use quantum effects to detect physical quantities, such as atomic magnetometer [1,2], atomic gyroscope [3,4], atomic clock [5], and superconducting quantum interferometer [6,7], are magnetic field-sensitive.

Point 2: In Introduction: Please replace "to attract magnetic lines of force into material" by "to cancel the magnetic flux density around the material"; Please replace "magnetic induction intensity" with "magnetic flux density"; This statement is unclear. What means "isolates" and what do you assume under "environment"?

Response 2: We thank for the reviewer’s comments. According to your suggestion, “to cancel the magnetic flux density around the material” are written instead of “to attract magnetic lines of force into material”; “magnetic flux density” are written instead of “magnetic induction intensity”; We are described this sentence in the manuscript for better understanding. This type of magnetic shield successfully suppresses the influence of the fluctuating magnetic field in the environment on the sensors inside the magnetic shield.

Page 1 Line 37: The passive magnetic shield uses the characteristics of higher permeability of magnetic material compared with air to cancel the magnetic flux density around the material.

Page 1 Line 38: Utilizing a magnetic material with a higher permeability sufficiently reduces the magnetic flux density within the magnetic shield.

Page 1 Line 41: This type of magnetic shield successfully suppresses the influence of the fluctuating magnetic field in the environment on the sensors inside the magnetic shield.

Point 3: Axial direction and radial direction should be written instead of z- direction and x-direction.

Response 3: We thank for the reviewer’s comments. According to your suggestion, “Axial direction and radial direction” are written instead of “z- direction and x-direction”.

Page 1 Line 41: In Figure 1, the x-direction is the radial, while the z-direction is the axial.

Point 4: In section 2.2, Please replace "magnetic induction intensity" by "magnetic flux.

Response 4: We thank for the reviewer’s comments. According to your suggestion, “magnetic flux density” are written instead of“magnetic induction intensity”.

Page 4 Line 128: The magnetic shielding coefficient is represented by the coefficient S. It is defined as the ratio of the magnetic flux density Bshield at center point of magnetic shield without the shield to the magnetic flux density B0 with the shield at the same point when the shield is present.

Point 5: In section 2.2, Please put a reference to ANSYS.

Response 5: We thank for the reviewer’s comments. According to your suggestion, we have added an introduction to the simulation software.

Page 4 Line 132: The magnetic shielding coefficient for complex structures can be calculated using FEM and the commercial program ANSYS Electromagnetics Suite 19.2 (ANSYS Maxwell 3D, Us).

Point 6: In section 2.2, How do you take into account the magnetic field of the Earth that is about 25-65 uT with some variations? Please skip an indent (Page 5 Line 151).

Response 6: We thank for the reviewer’s comments. The magnetic flux density is positioned at 25 μT because when we measure the environmental magnetic field, the maximum direction of the magnetic field is 25 μT, which we explained in the manuscript.

Page 4 Line 136: During the simulation, the magnetic flux density used is 25 μT, which is the actual measured magnetic flux density of the environment.

Page 5 Line 151: where  and  are the eddy current power loss and hysteresis power loss of ferrite shield, respectively. and  are the eddy current power loss and hysteresis power loss of mu-metal film, respectively.

Point 7: In section 3.1, Please replace "magnetic induction intensity" by "magnetic flux density"; Why do you have the air gaps?

Response 7: We thank for the reviewer’s comments. According to your suggestion, “magnetic flux density” are written instead of“magnetic induction intensity”; We added the source of air gap in the manuscript. Ferrite magnetic shield is difficult to manufacture and process, especially when large size is required. For this reason, a cylindrical ferrite shield is usually made of several short ferrite annuli pasted with ceramic adhesive. The thickness of the ceramic adhesive and the unevenness of the contact surface lead to air gaps.

Page 5 Line 156: Ferrite magnetic shield is difficult to manufacture and process, especially when large size is required. For this reason, a cylindrical ferrite shield is usually made of several short ferrite annuli pasted with ceramic adhesive. The thickness of the ceramic adhesive and the unevenness of the contact surface lead to air gaps [23].

Point 8: Line 166: Figure 4 should be written instead of Figure 3

Response 8: We thank for the reviewer’s comments. According to your suggestion, we have corrected the error. Figure 4 is written instead of Figure 3.

Page 5 Line 166: As shown in Figure 4, the x- and z-direction shielding coefficients decreased by 16.2% and 98.1%, respectively, when the gap width changed from 0 to 0.5 mm.

Point 9: In section 4, Please describe this abbreviation.

Response 9: We thank for the reviewer’s comments. According to your suggestion, we have added abbreviations.

Page 5 Line 166: The noise may be measured by building an ultra-high sensitive magnetic shielding measurement device based on the spin-exchange relaxation-free (SERF) effect.

Point 10: Since you have studied the influence of the alternative external magnetic field to the FCMS structure, you have to describe the frequency and the amplitude of the external field. In addition you have to study the influence of the signal frequency onto your shield efficiency.

Response 10: We thank for the reviewer’s comments. For MnZn ferrite materials, σ≈1 Ω-1m-1, correspondingly, the threshold frequency is higher than kHz, and below the threshold frequency, the shielding coefficient of ferrite magnetic shield does not change with the frequency, while the application frequency of SERF magnetometer is below 100 Hz. We also actually measured the frequency dependence of the shielding coefficient of the FMCS structure from DC to 100Hz. The change of the shielding coefficient with frequency is only 2%. We added this description to the manuscript.

Page 10 Line 280: For MnZn ferrite materials, σ≈1 Ω-1m-1, correspondingly, the threshold frequency is higher than kHz, and below the threshold frequency, the shielding coefficient of ferrite magnetic shield does not change with the frequency, while the application frequency of SERF magnetometer is below 100 Hz. We also actually measured the frequency dependence of the shielding coefficient of the FMCS structure from DC to 100 Hz. The change of the shielding coefficient with frequency is only 2%.

Reviewer 3 Report

This paper presents a magnetic shield made on a concentric combination of MnZn ferrite and a mu-metal film.

The paper is well written, with a nice introduction explaining the goal of the study and the interest of combining the MzZn ferrite with a mu-metal film. The theoritical method and the experiments are well described. The paper adresses at the same time novice readers (basic explanations are given) and experts in the field (valuable experimental are given). Thus it deserves to be published.

However, the paper may be improved by introducing the following minor corrections/addings

1) Line 166 : "As shown in Figure 4, the x- and z-direction shielding..." instead of "As shown in Figure 3, the x- and z-direction shielding..."

2) In section 2.2, the shielding coefficient is introduced as depending on the point it is measured. However, in section 3, it is never said at which point it is evaluated. I assume it is at the center of the cylindrical shield. Please mention it in the text.

3) For the results shown in Fig. 3, we do not know the mu-metal film thickness. Please, give it the value.

4) Lines 270-273 : the authors mentioned that the residual magnetic field inside the shield for z-direction shielding may exceed the measurng range of the atomic magnetometer. Then they give experimental results!!! So how were performed the measurements ?

Author Response

Response to Reviewer 3 Comments

We would first like to thank the reviewers for carefully reading our paper and giving us the opportunity to improve its quality. The revisions have been highlighted in the manuscript according to the requirement. Synchronous modifications and explanations have been carried out in the revised manuscript.

Point 1: Line 166: "As shown in Figure 4, the x- and z-direction shielding..." instead of "As shown in Figure 3, the x- and z-direction shielding..."

Response 1: We thank for the reviewer’s comments. According to your suggestion, we have corrected the error. Figure 4 is written instead of Figure 3.

Page 5 Line 166: As shown in Figure 4, the x- and z-direction shielding coefficients decreased by 16.2% and 98.1%, respectively, when the gap width changed from 0 to 0.5 mm.

Point 2: In section 2.2, the shielding coefficient is introduced as depending on the point it is measured. However, in section 3, it is never said at which point it is evaluated. I assume it is at the center of the cylindrical shield. Please mention it in the text.

Response 2: Thanks very much for your comments and guidance. It is defined as the ratio of the magnetic flux density Bshield at center point of magnetic shield when the shield is present to the magnetic flux density B0 with the shield at the same point without the shield.

Page 2 Line 13: It is defined as the ratio of the magnetic flux density Bshield at center point of magnetic shield when the shield is present to the magnetic flux density B0 with the shield at the same point without the shield.

Point 3: For the results shown in Fig. 3, we do not know the mu-metal film thickness. Please, give it the value.

Response 3: Thanks very much for your comments and guidance. We added the value of mu-metal film thickness. The thickness of mu-metal film used for simulation is 0.5mm.

Page 5 Line 163: The thickness of mu-metal film used for simulation is 0.5mm.

Point 4: Lines 270-273: the authors mentioned that the residual magnetic field inside the shield for z-direction shielding may exceed the measurng range of the atomic magnetometer. Then they give experimental results!!! So how were performed the measurements?

Response 4: Thanks very much for your comments and guidance. The external magnetic field applied when measuring the z-direction magnetic shielding coefficient of conventional magnetic shielding was between 50 and 120 nT, and the external magnetic field applied when measuring the x-direction magnetic shielding coefficient of conventional magnetic shielding was between 400 and 18000 nT. The reason why the magnetic field applied in the z-direction is smaller than that in the x-direction is to prevent the residual magnetic field inside the shield from exceeding the measuring range of the atomic magnetometer due to the smaller measuring range of the atomic magnetometer and the smaller shielding coefficient in the z-direction..

Page 10 Line 280: The external magnetic field applied when measuring the z-direction magnetic shielding coefficient of conventional magnetic shielding was between 50 and 120 nT. The reason why the magnetic field applied in the z-direction is smaller than that in the x-direction is to prevent the residual magnetic field inside the shield from exceeding the measuring range of the atomic magnetometer due to the smaller measuring range of the atomic magnetometer and the smaller shielding coefficient in the z-direction. 

We tried to satisfy the reviewers’ request at the best of our capabilities. Besides, we have reviewed the entire manuscript and double checked author affiliations, copyright, figures, references and the details in main text. We hope that our revised manuscript is now acceptable for a publication in Materials.

We look forward to your favorable decision.

Yours sincerely,

Danyue Ma et al.

Round 2

Reviewer 2 Report

Dear Authors,

Thanks for your corrections of the manuscript. Your article can be accepted for publication.